# Comparative characteristics of oat doubled haploids and oat × maize addition lines: Anatomical features of the leaves, chlorophyll *a* fluorescence and yield parameters

**Marzena Warchoł**[ID][1]*, **Katarzyna Juzoń-Sikora**[1], **Dragana Rančić**[ID][2], **Ilinka Pećinar**[2], **Tomasz Warzecha**[3], **Dominika Idziak-Helmcke**[4], **Kamila Laskoś**[1], **Ilona Czyczyło-Mysza**[1], **Kinga Dziurka**[1], **Edyta Skrzypek**[1]

**1** Polish Academy of Sciences, The Franciszek Górski Institute of Plant Physiology, Kraków, Poland, **2** Faculty of Agriculture, University of Belgrade, Belgrade, Serbia, **3** Department of Plant Breeding, Physiology and Seed Science, University of Agriculture in Kraków, Kraków, Poland, **4** Institute of Biology, Biotechnology, and Environmental Protection, University of Silesia in Katowice, Katowice, Poland

* m.warchol@ifr-pan.edu.pl

**Data Availability Statement:** All relevant data are within the manuscript and its Supporting information files.

## Abstract

As a result of oat (*Avena sativa* L.) × maize (*Zea mays* L.) crossing, maize chromosomes may not be completely eliminated at the early stages of embryogenesis, leading to the oat × maize addition (OMA) lines development. Introgression of maize chromosomes into oat genome can cause morphological and physiological modifications. The aim of the research was to evaluate the leaves' anatomy, chlorophyll *a* fluorescence, and yield parameter of oat doubled haploid (DH) and OMA lines obtained by oat × maize crossing. The present study examined two DH and two disomic OMA lines and revealed that they differ significantly in the majority of studied traits, apart from: the number of cells of the outer bundle sheath; light energy absorption; excitation energy trapped in PSII reaction centers; and energy dissipated from PSII. The OMA II line was characterized by larger size of single cells in the outer bundle sheath and greater number of seeds per plant among tested lines.

## Introduction

As a result of distant crossings of cereals, hybrid embryos of wheat, oat, and barley are formed, and during subsequent divisions of the nucleus, the paternal chromosomes are eliminated, which in turn leads to the formation of haploid embryos with only the mother's genetic material [1–3]. However, sometimes this elimination of paternal chromosomes does not occur properly and fragments or complete donor chromosomes are incorporated into the mother's genome. Most often it takes place in the crosses of plants belonging to the two subfamilies Pooideae and Panicoideae, within the family Poaceae [4]. Extra chromosomes of maize (*Zea mays* L.) or pearl millet (*Pennisetum glaucum* (L.) R. Br.) have been observed in both haploids of wheat [1, 5] and oat [6]. Studying the causes of this phenomenon, Mochida et al. [7] found incomplete attachment of the spindle to maize centromeres, and Ishii et al. [8] identified the

**Funding:** This research was funded by the F. Górski Institute of Plant Physiology Polish Academy of Sciences, project No T.1 ZB.2/2019, and by the Ministry of Science, Technological Development, and Innovation of the Republic of Serbia No 451-03-47/2023-01/200116. The oat DH and OMA lines were produced under the National Centre for Research and Development Poland grant no. PBS3/B8/17/2015. The funders did not and will not have a role in study design, data collection and analysis, decision to publish, or preparation of the manuscript.

**Competing interests:** The authors have declared that no competing interests exist.

breakage of pearl millet chromosomes. The presence of stable maize chromosomes in the oat genome was first described by Riera-Lizarazu et al. [6], and a few years later Kynast et al. [9] called them oat × maize addition (OMA) lines. Since then maize chromosome-addition lines of oat represent unique tools for researching the maize gene expression in oat, the structure of maize chromatin, or intergeneric translocations [6]. These lines have also been used for physical mapping of the maize genome [10], centromere-specific histone studies CENH3 [11], or the introduction of maize traits into oat, such as disease resistance [12]. The OMA lines, which are hybrids of $C_3$ and $C_4$ plants, were used by Kowles et al. [13] not only to better understood the $C_4$ photosynthesis mechanism but also for studying the possibilities of the transfer of $C_4$ photosynthesis from maize to oat. To analyze the genetics of the $C_4$ pathway in maize and to identify the chromosomes and chromosome areas that are crucial for $C_4$ photosynthesis, the expression of maize phosphoenolpyruvate carboxylase (*PEPc*) and pyruvate orthophosphate dikinase (*PPDK*) genes at the mRNA level in appropriate OMA lines was studied [13]. PEPc enzyme activity was discovered to be 5.0 times higher in the OMA9 line, and PPDK enzyme activity to be 17.6 times higher in the OMA6 line than that of the oat background line.

Dengler and Nelson [14] and Sage et al. [15] reported 22 versions of the two-celled Kranz anatomy, and according to these authors, five varieties of the Kranz anatomy were found in grasses that contained two sheaths encircling each vascular bundle. The parenchymal bundle sheath (PBS) surrounds the inner sheath, which is known as the mestome sheath and possesses endodermal features. The PBS is where Rubisco is located in each of the Kranz variations and is used to fix $CO_2$. To our knowledge, only Tolley et al. [16] conducted research on the anatomical structure of the leaves of the OMA lines, and showed that two key traits of $C_4$ photosynthesis, increased PBS cell size and reduced vein spacing in the $C_3$ leaf, can be changed by the introduction of maize chromosomes into the oat genome. Moreover, these authors studied lipid deposition in oat vascular bundle sheath cells, and indicated that chromosome 3 likely contains loci that can control PBS cell wall lipid deposition but not the entire suberin deposition pathway.

$C_4$ plants grow faster than $C_3$ plants and also require less water. Thus plant biologists would like to introduce certain $C_4$ traits into $C_3$ crop plants including tolerance to rapidly changing environmental factors [17]. Juzoń et al. [18] evaluated the characteristics of the OMA line under soil drought conditions and discovered that OMA lines outperform the Bingo cultivar with regard to of photosynthetic apparatus performance under both optimum soil water content and unfavorable conditions. Recently, Warzecha et al. [19] stated that the effect of drought stress was significant in all considered parameters of chlorophyll *a* fluorescence kinetics (PCF). The overall performance of PSII photochemistry (PI) and the pool size of the electron acceptors (Area), were the parameters of the PSII photosystem that were most impacted by drought. Because drought is considered as one of the most important factors limiting the productivity of crops, transgenic methods to modify $C_4$ photosynthetic enzymes in an effort to convert $C_3$ plants into $C_4$ plants is a current field of research. In order to increase yield potential, engineering $C_4$ photosynthesis into rice is being attempted by the international $C_4$ Rice Consortium [20].

Many of the OMA lines display specific phenotypes, which indicates that maize genes are likely expressed and capable of altering the phenotype of oat plants [21]. These findings were corroborated by the work of Skrzypek et al. [22], in which the OMA lines differed from the doubled haploid (DH) lines. Moreover, the earlier study of Kynast et al. [9] revealed that the presence of maize chromosomes in the oat genome caused an erect leaf phenotype, or the appearance of leaf blades with necrotic and chlorotic spots compared to *cv*. Sun II. This founding is contrary to Warzecha et al. [23] study, were all fifteen OMA lines were oat-shaped plants.

The aim of our study was to evaluate how the presence of maize chromosomes changes the anatomical parameters of the leaves and functioning of photosynthetic apparatus of the disomic OMA lines compared to the DH lines and Bingo cultivar. We also assessed yield components for all lines. This research extends knowledge about hybrids in the context of more targeted efforts to improve crop productivity. To our knowledge, for the first time, the OMA and DH lines produced from the same parental crosses were compared in terms of leaf anatomical features, taking into account the parameters of chlorophyll *a* fluorescence kinetics and yielding. We would like to show that the differences between the tested plants may result not only from the presence or absence of maize chromosomes in the oat genome, but are also the result of parental background.

## Results

### Identification of oat × maize hybrids

The presence of maize chromosome introgression in OMA I and OMA II, which was detected by PCR using primers specific for ubiquitous highly repetitive maize retrotransposon Grande-1 (Fig 1A, S1 Raw image), was further validated by GISH using maize gDNA as a probe. The experiments did not require using unlabeled oat genomic DNA in order to block non-specific hybridization between oat and maize due to high phylogenetic distance between those species. In both OMA lines GISH clearly detected the presence of two maize chromosomes added to a full set of oat chromosomes. In contrast, DH I and DH II both possessed only a clear set of *Avena sativa* L. chromosomes (Table 1, Fig 1B). The analysis with maize chromosome-specific SSR markers revealed that the OMA I line was a disomic characterized by the presence of a pair of maize chromosome 5 (2n = 44 (42 + Zm5")) (Fig 1C), whereas line OMA II was a double monosomic with maize chromosomes 3 and 8 (2n = 44 (42 + Zm3' + Zm8')).

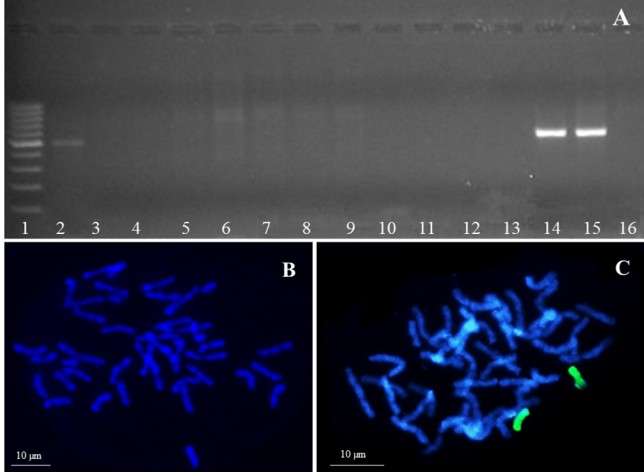

**Fig 1. Identification of maize (*Zea maize* L.) chromatin added to the oat (*Avena sativa* L.) genome by PCR and genomic *in situ* hybridization (GISH).** (A) The agarose gel with bands representing DNA fragments *Grande 1* (500 bp) specific for maize; path 1 –marker leader, path 2 –maize cv. Waza, path 3 –oat cv. Bingo, paths 4–13 DH lines of oat, paths 14–15 OMA lines, path 16 DH line of oat (B) Chromosomes of doubled haploid line (DH I), (C) Chromosomes of oat × maize addition line (OMA I). Blue fluorescence: DAPI, green fluorescence: maize genomic DNA (gDNA).

**Table 1. Characteristics of the plant material used in the study.**

| Line/cultivar | Origin | No. of maize chromosome added to oat genome | Chromosome ID |
|---|---|---|---|
| DH I | Flamingstern × Bingo | - | - |
| OMA I | Flamingstern × Bingo | 2 | Zm5" |
| DH II | STH 9787(b) × Bingo | - | - |
| OMA II | STH 9787(b) × Bingo | 2 | Zm3' + Zm8' |
| *cv.* Bingo | - | - | - |

## Anatomical analysis of the DH and OMA leaves

An analysis of the variance showed significant differences between tested oat plants in terms of all the tested anatomical features (Table 2). In order to investigate whether the presence of individual maize chromosomes increased the number of vascular bundles in oat, distances between the midpoints of adjacent vascular bundles were measured in transverse sections of leaves from DH and OMA lines and oat cv. Bingo (Fig 2A–2E). In the DH I and OMA I lines, which were both developed from Flamingstern × Bingo crosses, the mean distances between adjacent vascular bundles were approximately 400 μm and were bigger than in the DH II and OMA II lines (STH 9787(b) × Bingo) and in cv. Bingo (341 μm, 353 μm, and 334 μm, respectively) (Table 3). The DH I line was characterized by having the largest mesophyll area among all the tested oat plants and amounted to 86,791.12 μm$^2$, while the OMA I line and the cv. Bingo had similar values of the examined parameter (75,881.07 μm$^2$; 70,159.06 μm$^2$, respectively). We found that the distance between the two bundles was affected by the germplasm used in the crosses and was not dependent on maize chromosomal introgression. Although there were no statistically significant differences in the number of cells of the outer bundle sheath, the entire area of the bundle sheath differed among the tested oat plants from the largest in OMA I and DH I (6670.36 μm$^2$, 6062.36 μm$^2$, respectively), to the smallest in OMA II and cv. Bingo (3711.20 μm$^2$, 3972.04 μm$^2$, respectively). In this study, attention was paid to the size of the cells forming the bundle sheath. The biggest size of a single cell in the outer bundle sheath, more than 500 μm$^2$, was noted in DH I and OMA I lines (Fig 2F).

**Table 2. One-way analysis of variance for leaf anatomical traits of DH and OMA lines and *cv.* Bingo.**

| Trait | SS | df | MS | F |
|---|---|---|---|---|
| Distance between two bundles [μm] | 7.058 | 4 | 1.764E+04 | 6.351*** |
| Area of mesophyll between two bundles [μm$^2$] | 5.153 | 4 | 1.288E+09 | 19.260*** |
| Number of cells of inner bundle sheath | 3.272 | 4 | 8.180E+00 | 3.343** |
| Area of inner bundle sheath [μm$^2$] | 2.734 | 4 | 6.834E+05 | 4.136** |
| Average size of single cell in inner bundle sheath [μm$^2$] | 1.205 | 4 | 3.012E+03 | 4.753** |
| Number of cells of outer bundle sheath | 4.520 | 4 | 1.130E+00 | 0.776[ns] |
| Area of outer bundle sheath [μm$^2$] | 6.647 | 4 | 1.662E+07 | 22.018*** |
| Average size of single cell in outer bundle sheath [μm$^2$] | 6.086 | 4 | 1.521E+05 | 25.809*** |

SS—sum of squares; df—degrees of freedom; MS—mean square; *F*—F-test;

* $p \leq 0.05$,

** $p \leq 0.01$,

*** $p \leq 0.001$,

ns—not significant

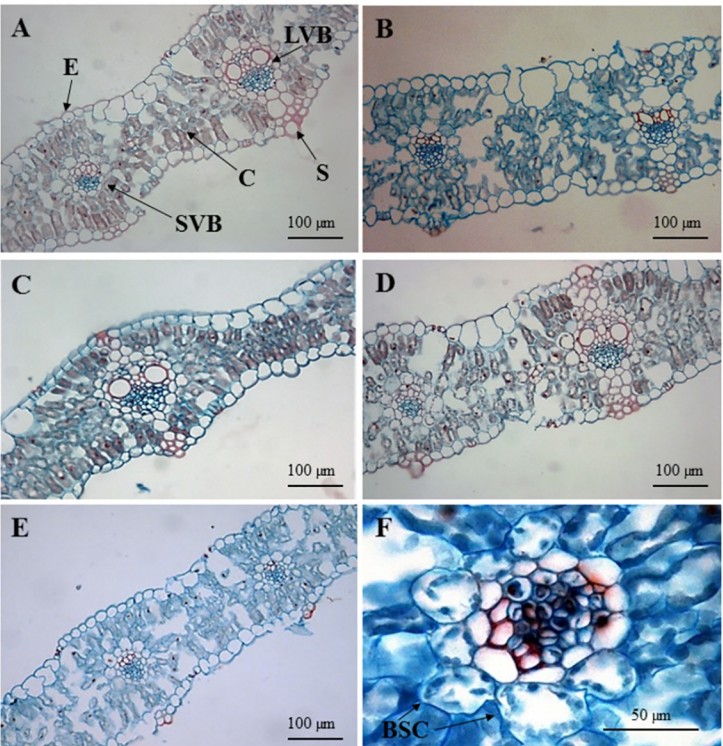

**Fig 2.** Cross sections of oat (*Avena sativa* L.) leaves made by using the IM 1000 software package: (A) doubled haploid (DH I); (B) oat × maize addition line (OMA I); (C) doubled haploid (DH II); (D) oat × maize addition line (OMA II); (E) oat *cv.* Bingo; (F) vascular bundle of oat × maize addition line (OMA II); LVB–large vascular bundle, SVB–small vascular bundle, C–chlorenchyma, BSC–bundle sheath cells, S–sclerenchyma, E–epidermis.

## Characteristics of the photosynthetic apparatus of DH and OMA lines

Selected PCF were chosen in order to compare the functioning of photosynthetic apparatuses of the DH and OMA lines. The analysis of variance presented in Table 4 showed significant differences between tested oat plants in terms of Fv/Fm (maximum photochemical efficiency

**Table 3. Leaf anatomical traits of DH and OMA lines and *cv.* Bingo.** The mean values (n = 5) ± SE marked with different letters are significantly different at p ≤ 0.05 according to the Duncan's multiple test.

| Line/ Cultivar | Distance between two small bundles [µm] | Area of mesophyll between two bundles [µm²] | Number of cells of inner bundle sheath | Area of inner bundle sheath [µm²] | Average size of single cell in inner bundle sheath [µm²] | Number of cells of outer bundle sheath | Area of outer bundle sheath [µm²] | Average size of single cell in outer bundle sheath [µm²] |
|---|---|---|---|---|---|---|---|---|
| DH I | 427.81 ± 10.54 a | 86791.12 ± 2491.94 a | 11.70 ± 0.27 b | 1698.78 ± 61.74 a | 126.90 ± 4.34 a | 11.60 ± 0.31 a | 6062.36 ± 314.32 a | 542.96 ± 32.03 a |
| OMA I | 407.19 ± 16.45 a | 75881.07 ± 773.49 b | 12.60 ± 0.57 ab | 1735.74 ± 157.56 a | 124.37 ± 9.34 a | 12.10 ± 0.31 a | 6670.36 ± 300.58 a | 548.40 ± 20.85 a |
| DH II | 341.25 ± 15.81 b | 61126.58 ± 3216.08 c | 13.40 ± 0.40 a | 1356.16 ± 131.25 ab | 115.27 ± 9.13 ab | 11.80 ± 0.61 a | 3711.20 ± 217.70 c | 266.79 ± 17.47 c |
| OMA II | 353.49 ± 14.51 b | 58968.41 ± 2738.78 c | 13.80 ± 0.71 a | 1338.22 ± 131.11 ab | 95.30 ± 5.97 bc | 12.30 ± 0.21 a | 4843.76 ± 154.66 b | 401.40 ± 19.03 b |
| *cv.* Bingo | 334.32 ± 23.43 b | 70159.06 ± 2972.86 b | 13.80 ± 0.40 a | 1120.56 ± 140.31 b | 88.35 ± 9.61 c | 11.50 ± 0.34 a | 3972.04 ± 342.73 c | 346.24 ± 28.70 b |

**Table 4. One-way analysis of variance for chlorophyll *a* fluorescence kinetics parameters of DH and OMA lines and *cv*. Bingo.** Fv/Fm—maximum photochemical efficiency of PSII; Area—area over the chlorophyll a fluorescence induction curve; PI—overall performance index of PSII photochemistry; ABS/CS—light energy absorption; $TR_0/CS$—excitation energy trapped in PSII reaction centers; $ET_0/CS$—energy used for electron transport; $DI_0/CS$—energy dissipated from PSII; $RC/CS_0$—number of active reaction centers.

| Trait | SS | Df | MS | *F* |
|---|---|---|---|---|
| Fv/Fm | 2,055E-03 | 4 | 0.001 | 5.611** |
| Area | 1,553E+09 | 4 | 388353600.00 | 6.461*** |
| P.I. | 2,067E+01 | 4 | 5.168 | 4.390** |
| ABS/CS | 1,626E+02 | 4 | 40.640 | 0.196[ns] |
| $TR_0/CS$ | 3,297E+02 | 4 | 82.424 | 0.610[ns] |
| $ET_0/CS$ | 7,044E+02 | 4 | 176.110 | 3.137* |
| $DI_0/CS$ | 2,099E+02 | 4 | 52.485 | 2.695[ns] |
| $RC/CS_0$ | 1,397E+03 | 4 | 349.159 | 2.975*** |

SS—sum of squares; df—degrees of freedom; MS—mean square; *F*—F-test;

* $p \leq 0.05$,

** $p \leq 0.01$,

*** $p \leq 0.001$,

ns—not significant

of PSII), Area (area over the chlorophyll *a* fluorescence induction curve), PI (overall performance index of PSII photochemistry) $ET_0/CS$ (energy used for electron transport) and $RC/CS_0$ (number of active reaction centers in the excited leaf fragment) as opposed to ABS/CS (light energy absorption), $TR_0/CS$ (excitation energy trapped in PSII reaction centers), and $DI_0/CS$ (energy dissipated from PSII). Data presented in Fig 3 showed that the OMA I line usually achieved the highest PCF indices apart from energy dissipation from PSII ($DI_0/CS$) compared to the other oat plants. The individuals from this line were characterized by maximum photochemical efficiency of PSII (Fv/Fm) and the biggest area over the chlorophyll *a* fluorescence induction curve (Area), which corresponded to the size of the reduced plastoquinone pool. Also, the overall performance index of PSII photochemistry (PI) was much higher than in other plants. The OMA I line plants used more energy for electron transport ($ET_0/CS$) but they were not the only ones with a high number of active reaction centers in the excited leaf fragment ($RC/CS_0$). The maximum photochemical efficiency of PSII was achieved by the OMA I line although the values of this parameter did not differ significantly in DH I and cv. Bingo. Photosynthetic apparatuses of all lines showed the same level of light energy absorption (ABS/CS), as well as excitation energy trapped in PSII reaction centers ($TR_0/CS$).

## Morphology and yield components of DH and OMA lines

An analysis of the variance revealed significant differences between tested oat plants in terms of all morphological traits and yield components (Table 5). The tested lines showed differences in flag leaf morphology, with the longest flag leaf observed in *cv*. Bingo (25.67 cm), and the shortest in OMA I (14.33 cm) (Fig 4). Despite this, the flag leaf of the latter line was the widest (9.33 mm). The OMA I line also had the tallest main shoot and the longest panicle, while those morphological traits were the smallest in the DH II and OMA II lines. The highest production of lateral shoots (9.00) as well as the number (262.67) and weight (16.49 g) of seeds per plant was found in *cv*. Bingo. On the other hand, no significant differences regarding the above mentioned yield components were observed between all tested lines. The results concerning TGW were particularly notable. The highest value of the TGW was recorded for cv. Bingo and for

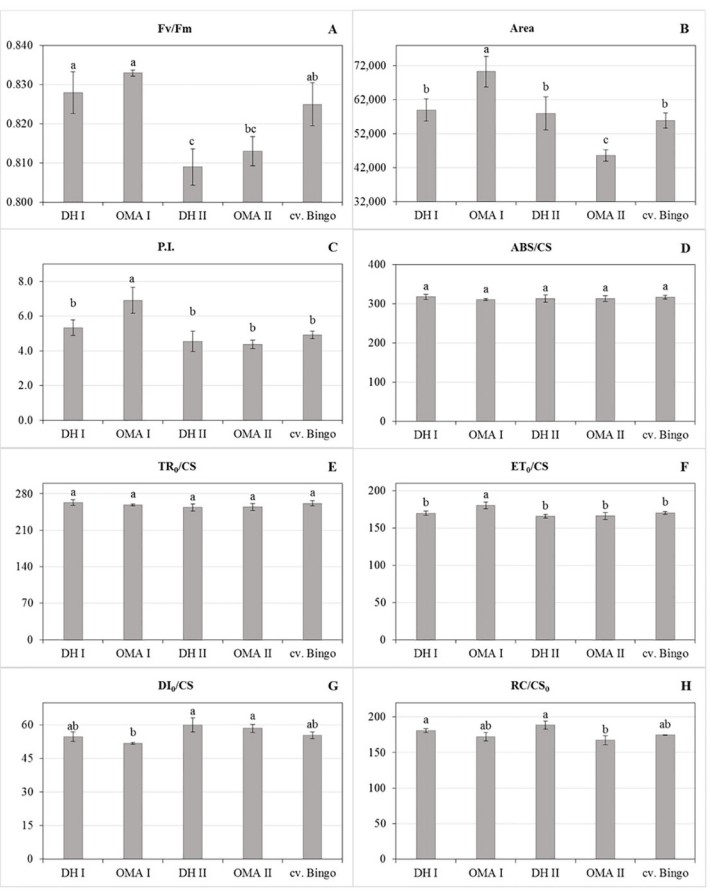

**Fig 3. Chlorophyll *a* fluorescence kinetics parameters of the DH and OMA lines and cv. Bingo leaves.** (A) Fv/Fm—maximum photochemical efficiency of PSII; (B) Area—area over the chlorophyll a fluorescence induction curve; (C) PI—overall performance index of PSII photochemistry; (D) ABS/CS—light energy absorption; (E) $TR_0/CS$—excitation energy trapped in PSII reaction centers; (F) $ET_0/CS$—energy used for electron transport; (G) $DI_0/CS$—energy dissipated from PSII; (H) $RC/CS_0$—number of active reaction centers in the excited leaf fragment. The mean values (n = 5) ± SE marked with different letters are significantly different at $p \leq 0.05$ according to the Duncan's multiple test.

**Table 5. One-way analysis of variance for morphological traits and yield components of DH and OMA lines and *cv*. Bingo.**

| Source of variation | SS | df | MS | F |
|---|---|---|---|---|
| Flag leaf length [cm] | 256.267 | 4 | 64.067 | 3.852* |
| Flag leaf width [mm] | 50.267 | 4 | 12.567 | 9.921*** |
| Height of main shoot [cm] | 3478.933 | 4 | 869.733 | 51.924*** |
| Panicle length [cm] | 75.233 | 4 | 18.808 | 22.127*** |
| Number of shoots | 23.600 | 4 | 5.900 | 9.833*** |
| Number of seeds/plant | 108488.933 | 4 | 27122.233 | 17.120*** |
| Weight of seeds/plant [g] | 509.828 | 4 | 127.457 | 19.350*** |
| Thousand grain weight (TGW) [g] | 1942.521 | 4 | 485.630 | 22.229*** |

SS—sum of squares; df—degrees of freedom; MS—mean square; *F*—F-test;

* $p \leq 0.05$,

** $p \leq 0.01$,

*** $p \leq 0.001$,

ns—not significant

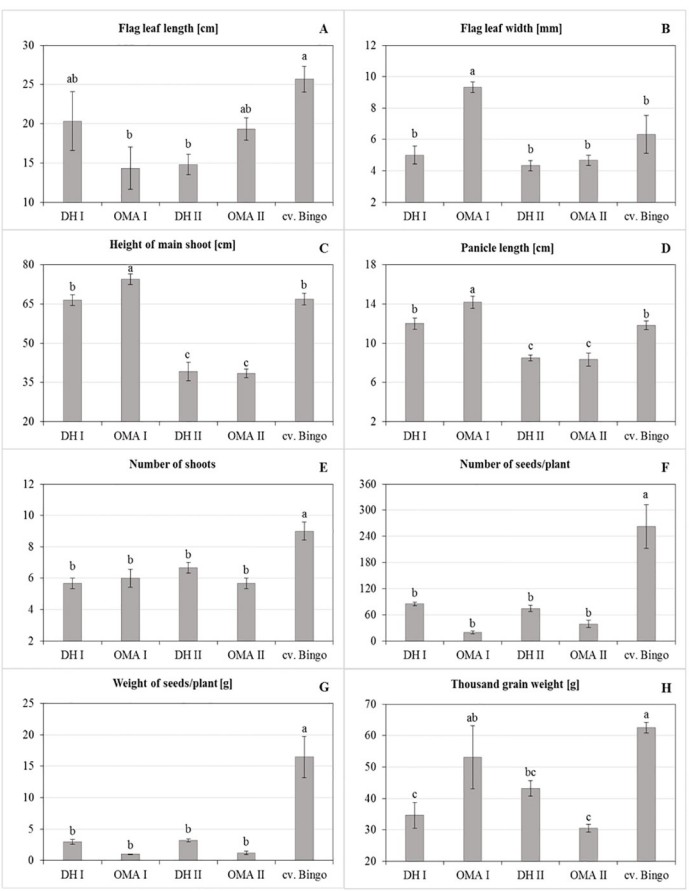

**Fig 4.** The yield components of the DH and OMA lines and *cv*. Bingo: (A) flag leaf length [cm]; (B) flag leaf width [mm]; (C) height of the main shoot [cm]; (D) panicle length [cm]; (E) number of shoots; (F) number of seeds per plant; (G) weight of seeds per plant [g]; (H) thousand grain weight (TGW) [g]. The mean values (n = 5) ± SE marked with different letters are significantly different at p ≤ 0.05 according to the Duncan's multiple test.

the OMA I line (62.50 g; 53.10 g, respectively). Additionally, OMA I produced the smallest number of seeds per plant (0.99) among all tested lines.

## Biplot analysis and correlations among yield components and chlorophyll *a* fluorescence parameters

The oat genotypes in biplot analysis were compared for all yield traits and chlorophyll *a* fluorescence parameters (Fig 5). A straight angle indicates no correlation, an obtuse angle indicates a negative correlation, and an acute angle indicates a positive correlation between the measured data. Principal component analysis (PCA) showed that yield components as well as chlorophyll *a* fluorescence parameters separated the oat genotypes. Taking into account yield components, the first two principal components included 89.95% of the total variation (Fig 5A), whereas chlorophyll *a* fluorescence parameters 93.56% (Fig 5C). PCA revealed that the first PC explained 55.71% of the variation with measured yield components and 68.36% for chlorophyll *a* fluorescence parameters. The second PC explained 34.24% and 25.20% of the total variability. Thus, the *cv*. Bingo with lower PC1 and PC2 was superior in terms of yield parameters (Fig 5B) and in terms of chlorophyll *a* fluorescence parameters DH II and OMA II

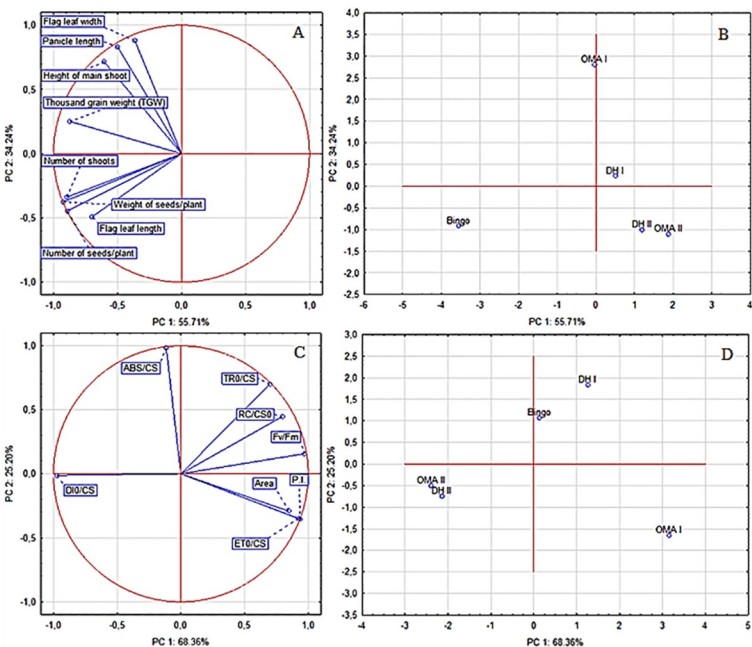

**Fig 5.** Biplot based on first two principal components axes (PC1 and PC2) for chlorophyll a fluorescence parameters of oat cv. Bingo, OMA and DH lines (A); distribution of oat genotypes based on the chlorophyll fluorescence parameters (B); for yield components of oat *cv*. Bingo, OMA and DH lines (C); distribution of oat genotypes based on the yield components (D).

(Fig 5D). The lines were grouped separately according to their origin, not number of retained maize chromosomes (Fig 5B and 5D).

Correlations coefficients between studied yield traits and chlorophyll *a* fluorescence parameters for all studied genotypes are presented in Table 6. The statistically significant correlations between height of main shoot and panicle length (r = 0.982), number of shoots and number (r = 0.936) and weight (r = 0.966) of seeds as well as number and weight of seeds per plant (r = 0.991) were found. Also some chlorophyll *a* fluorescence parameters showed significant correlations with yield parameters. Chlorophyll *a* fluorescence parameters Fv/Fm and $DI_0/CS$ shown high positive correlations with height of main shoot (r = 0.978 and r = 0.978, respectively), whereas high negative correlations with panicle length (r = -0.951 and r = -0.980, respectively). Moreover, high positive correlations between P.I. and $ET_0/CS$ and flag leaf width and panicle length (r = 0.966 and r = 0.966, respectively) were noted.

## Discussion

Interspecific plant crossings may omit fertilization obstacles and produce allopolyploids or haploids. Allopolyploids combine the genomes of both parental gametes, whereas haploids delete one parental genome upon fertilization. Chromosome addition lines, which fall between allopolyploids and haploids in terms of genome interaction, can be produced through certain uncommon hybridization occurrences [24]. In interspecific crossings, chromosome addition lines are typically kept and they include a full set of uniparental chromosomes along with one or more pairs of chromosomes that were stably inherited from the distinct parent [25]. Oat × maize addition (OMA) lines described by Riera-Lizarazu et al. [6] were created through a sexual cross between oat (*Avena sativa* L., 2n = 6x = 42) and maize (*Zea mays* L., 2n = 20),

**Table 6. Pearson coefficients of linear correlation (p ≤ .05) between chlorophyll *a* fluorescence parameters and yield components of oat plants.**

| Parameter | FLL | FLW | HMS | PL | NS | NS/P | WS/P | TGW | Fv/Fm | Area | P.I. | ABS/CS | TR$_0$/CS | ET$_0$/CS | DI$_0$/CS |
|---|---|---|---|---|---|---|---|---|---|---|---|---|---|---|---|
| FLW | -0.217 | | | | | | | | | | | | | | |
| HMS | 0.205 | 0.766 | | | | | | | | | | | | | |
| PL | 0.034 | 0.856 | **0.982***| | | | | | | | | | | | |
| NS | 0.655 | 0.074 | 0.213 | 0.105 | | | | | | | | | | | |
| NS/P | 0.852 | -0.097 | 0.249 | 0.094 | **0.936*** | | | | | | | | | | |
| WS/P | 0.820 | 0.010 | 0.285 | 0.144 | **0.966*** | **0.991*** | | | | | | | | | |
| TGW | 0.278 | 0.606 | 0.577 | 0.552 | 0.816 | 0.644 | 0.723 | | | | | | | | |
| Fv/Fm | 0.147 | 0.782 | **0.978*** | **0.978*** | 0.043 | 0.098 | 0.134 | 0.433 | | | | | | | |
| Area | -0.454 | 0.774 | 0.729 | 0.810 | -0.037 | -0.173 | -0.123 | 0.500 | 0.688 | | | | | | |
| P.I. | -0.389 | **0.922*** | 0.799 | **0.897*** | -0.198 | -0.305 | -0.233 | 0.383 | 0.836 | **0.895*** | | | | | |
| ABS/CS | 0.776 | -0.521 | 0.124 | -0.058 | 0.383 | 0.671 | 0.568 | -0.022 | 0.049 | -0.334 | -0.455 | | | | |
| TR$_0$/CS | 0.571 | 0.325 | 0.846 | 0.741 | 0.261 | 0.468 | 0.439 | 0.335 | 0.821 | 0.345 | 0.399 | 0.610 | | | |
| ET$_0$/CS | -0.261 | **0.976*** | 0.828 | **0.916*** | -0.061 | -0.182 | -0.095 | 0.496 | 0.859 | 0.837 | **0.980*** | -0.452 | 0.423 | | |
| DI$_0$/CS | -0.032 | -0.866 | **-0.951*** | **-0.980*** | 0.011 | 0.005 | -0.050 | -0.439 | **-0.986*** | -0.727 | **-0.901*** | 0.116 | -0.716 | **-0.927*** | |
| RC/CS$_0$ | -0.261 | -0.400 | -0.165 | -0.193 | 0.074 | 0.074 | 0.001 | -0.001 | -0.293 | 0.243 | -0.192 | 0.292 | -0.081 | -0.325 | 0.359 |

* $p < 0.05$

FLL—flag leaf length [cm], FLW—flag leaf width [mm], HMS—height of main shoot [cm], PL—panicle length [cm], NS—number of shoots, NS/P—number of seeds/plant, WS/P—weight of seeds/plant [g], TGW—thousand grain weight [g]

which resulted in adding specific maize chromosomes to the entire oat genome. In the oat genome background, a single maize chromosome makes up approximately 2% of the total nuclear DNA [26]. In general, hybrids with one or two maize chromosomes resemble oat plants at every stage of growth, making it a challenge to determine how many maize genes are expressed or silenced. Recently, Dong et al. [27] discovered that more than 70% of the alien maize chromosomes' genes kept their original pattern of expression or transcription in the oat genomic environment. They hypothesized that a local cis region in the foreign maize chromosome may be primarily responsible for controlling the maize genes preserving the original transcription in OMA lines. Just two studies, as far as we are aware, looked at chromosomal-wide gene expression in chromosome addition lines [27, 28].

The suppression of the $C_4$ pathway in hybrids between closely related $C_3$ and $C_4$ species has been documented by Brown and Bouton [29]. As researchers have shown, even the presence of a full genome complement fails to activate a $C_4$-like state in $C_3$–$C_4$ hybrids, which is why it is unlikely that an individual chromosome or chromosomes can initiate the $C_4$ program within the $C_3$ context. However, Kowles et al. [13] indicate that OMA lines possess entire chromosomes that contain maize transcription factors and promoters rather than just individual genes, which theoretically might lead to a rise in the activity of $C_4$ enzymes. Therefore, it seems reasonable to conduct research on the OMA lines that have the potential to help clarify the transfer of $C_4$ photosynthesis from maize to oat. The individual OMA lines were used to dissect the genetics of the $C_4$ pathway of maize and to identify maize chromosomes and chromosome regions that are important in $C_4$ photosynthesis [13]. While the mono- and dicotyledonous $C_4$ lineages' leaf design can vary significantly, all $C_4$ plants have a wreath-like arrangement of mesophyll and bundle sheath cells around the vascular bundles called the Kranz anatomy [14]. A dense vein spacing that is part of the Kranz anatomy causes a reduction in the volume of mesophyll relative to bundle sheath tissue, which in $C_4$ grasses can reach a ratio of almost one to one. The narrow vein spacing in $C_4$ species seems to be primarily caused

by modifications to the minor vascular bundle's patterning, rather than the major vascular bundle [30]. In our study after measuring the distance between the two small bundles, we did not observe a denser system of vascular bundle in either OMA line. However, Tolley et al. [16] showed that maize chromosomes are capable of altering vascular bundle spacing in $C_3$ leaves, but only one of two OMA lines in that study showed decreasing vascular bundle-compared with the wild type. Maize chromosome 9 was common to both tested OMA lines, but they differed in their parental backgrounds. The loci on maize chromosome 9 encoding this trait were silenced, depending on their interaction with the oat genome. Tolley et al. [16] reported that the lines possessing maize chromosome 1, 2, 3, 7, 8, or 9 had significantly larger PBS areas than either of the oat parental lines. This suggests that these chromosomal additions independently specified larger PBS cells. In our study, the area of the outer bundle sheath was the largest in line OMA I, which is in accordance with the above-cited studies, if it wasn't for that in line DH I (produced from the same crossing) this area was comparable with OMA I line. However, it is worth emphasizing that the OMA II line was characterized by a larger area of the outer bundle sheath not only in relation to the DH II line, but also to cv. Bingo. Overall, Dengler et al. [31] found that the pattern of variation in leaf blade anatomy is complex, reflecting correlations with both taxonomic group and photosynthetic type, and no new analytical tools have been developed to distinguish one biochemical type from another *a priori*.

The morphological observations of the examined plants revealed several differences within the DH and OMA lines. The OMA I line had taller main shoots, longer panicles, and shorter and wider flag leaf compared to the DH I line. Kynast et al. [9] also observed differences in hybrid morphology, and their OMA plants either produced necrotic and chlorotic patches on the leaf blades or were characterized by premature senescence that caused yellowing and drying of shoots. Similar findings were made by Riera-Lizarazu et al. [6], who noted that DH oat plants were more robust and viable than plants with maize chromosomes. Additionally, Juzoń et al. [18] demonstrated phenotypic variations within OMA plants that were produced by crossing the same parents and had four maize chromosomes. There were oat-like plants and grass-like plants among them, the former having tall stems that produced panicles, the latter not. The interaction of the oat and maize transcriptomes may have altered the expression of phenotype-linked oat genes, or some ectopic expression of phenotype-linked maize genes may have occurred within the genomic context of oat, leading to these modifications of phenotypes. In our research, we did not observe symptoms similar to those reported by other researchers; thus, the second hypothesis seems more plausible. Additionally, Dong et al. [27] stated that gain-of-function abnormalities are uncommon and that transcriptome analysis did not reveal any apparent enhanced expression of phenotype-related genes. Even though the oat genome has not yet been sequenced, further research on the impact of alien maize chromosomes on the oat transcriptome in OMA lines may provide information about the mechanisms behind the OMA phenotypes.

Interest in introducing the $C_4$ route into $C_3$ crops to boost yield potential is dictated by the $C_4$ plants higher productivity [15]. Due to higher photosynthetically active radiation-use efficiency, species such as maize, sorghum, and sugarcane can have 50% higher yields than $C_3$ species [32, 33]. Ways to increase the yield potential include improving photosynthesis and elevating photosynthetic utilization. It is worth stressing that in this study OMA lines were characterized by higher values of most of the analyzed PCF, which may confirm more efficient functioning of their photosynthetic apparatus compared to the DH lines and even the *cv.* Bingo. Similar observations were reported by Juzoń et al. [18], where among the examined OMA lines, the line with two retained maize chromosomes (2n = 44 (42 + Zm5″)) showed higher values for most parameters of chlorophyll a fluorescence. The greatest difference was observed in the Area, and this line showed almost two times higher values of this parameter.

The authors also emphasized that not only the number of retained maize chromosomes but also the identity of introduced maize chromosomes as well as oat genome are important in photosynthetic efficiency.

Our study on morphological features and yielding of oat confirmed the high rank of *cv*. Bingo, which was used for producing all tested DH and OMA lines. *Cv*. Bingo, as a model cultivar, was characterized by the longest flag leaf, the greatest production of lateral shoots, and the highest TGW. In turn, among the tested lines, OMA I also deserves attention, which, apart from the wider flag leaf, the tallest main shoot, and the longest panicle, had a TGW comparable to *cv*. Bingo. Similar findings were reported by Warzecha et al. [19], who found that under appropriate soil moisture, two OMA lines (9 and 78b) showed high producing potential comparable to cv. Bingo. Additionally, the lines previously stated showed greater yielding capability even under soil drought stress, which was reflected in a considerably lower decline in the quantity and mass of grains per plant. The researchers discovered that parental crossings rather than the introgression of maize chromosomes appear to be the primary cause of variations in seed production amongst examined lines. Literature records that, the evolutionary transition from $C_3$ ancestors to $C_4$ plants took place repeatedly and independently in a variety of taxonomically diverse groups [34, 35]. This could have led to a number of intermediates that were either matched or unmatched between anatomical features and physiological traits. Since already a few independent reversals from $C_4$ to $C_3$ have been reported [36], therefore, it seems probable that the ancestor of the Pooideae, to which oat, barley, and wheat belong, underwent a $C_4$ to $C_3$ reversal early in its evolution [37].

## Conclusions

The hypothesis that the introgression of maize chromosomes into the oat genome may cause changes in the leaf anatomy, plant morphology, photosynthetic efficiency, and yield was confirmed with regard to the following morphological features: flag leaf width, height of main shoot, panicle length, and weight of a thousand grains per plant in OMA I, especially when comparing with oat *cv*. Bingo. The introgression of maize chromosomes into the oat genome did not show comprehensive changes in leaf anatomy consistent with $C_4$ photosynthesis, but in the OMA I line the big cells of the outer bundle sheath and their size of about 500 $\mu m^2$ was similar to the size of these cells in maize. These data show that maize chromosomes are capable of modifying the size of the bundle sheath cells, the trait associated with $C_4$ photosynthesis, but this is definitely dependent on the oat genome.

## Materials and methods

### Plant material and growth conditions

The objects of our study were oat plants (*Avena sativa* L.): *cv*. Bingo; two DH lines (Flamingstern × Bingo and STH 9787(b) × Bingo); and two OMA lines (Flamingstern × Bingo and STH 9787(b) × Bingo) with two maize chromosomes (two Zm5" and one Zm3' + one Zm8', respectively) added to oat genome. Oat DH lines were obtained through oat × maize crosses according to the method described by Warchoł et al. [38]. Oat seeds were derived from Strzelce Plant Breeding Ltd., Małopolska Plant Breeding Polanowice Ltd. and Danko Plant Breeding Ltd. The day after pollinating emasculated oat flowers by maize, one drop of 100 mg $dm^{-3}$ 2,4-dichlorophenoxyacetic acid (2,4-D) was applied to each oat pistil. Developed embryos were isolated from ovaries 21 days after pollination and cultured on 190–2 medium [39], after which developed haploid plants were transferred on MS medium [40]. The embryos germinated at 21 ± 2 ˚C and light intensity equal to 60 $\mu mol\ m^{-2}\ s^{-1}$ and 16/8 h light/dark cycle. Haploid plants were acclimated to ex vitro conditions by transferring them to a moist

perlite and then to the soil at 20 ± 2 ˚C and light intensity 110 μmol m$^{-2}$ s$^{-1}$. After colchicine treatment, the oat plants were grown in a greenhouse in natural (solar) light intensity of 800 μmol m$^{-2}$ s$^{-1}$ during the day. Additional light was provided by sodium lamps between 6–8 a.m., 6–10 p.m., and on cloudy days at 21-28/17 ˚C day/night.

## Identification of oat × maize hybrids

DNA extraction (Genomic Mini AX Plant Kit, A&A Biotechnology, Gdynia, Poland) and PCR analyses (2720 Thermal Cycler; Applied Biosystems, Foster City, CA, USA) were performed according to Skrzypek et al. [22]. Two Grande-1F (50-AAA GAC CTC ACG AAA GGC CCA AGG-30), Grande-1R (50-AAA TGG TTC ATG CCG ATT GCA CG-30) primers (GenBank accession number X97604 [41]) were used for the PCR reaction, which in successive cycles enabled the amplification of the 500 bp retrotransposon region of *Grande-1* and detection of the presence of maize chromatin in oat plants. The obtained products were separated in 1.5% agarose gel with ethidium bromide (Sigma-Aldrich, St. Louis, MO, USA) in TBE buffer, under 90 V for 90 min. DNA markers of 100 bp to 1000 bp and concentration of 0.5 mg/ml (GeneRuler 100bp; Fermentas, Waltham, MA, USA) were used to estimate the length of PCR products. The image of electrophoretic separation was archived using the Imagemaster VDS gel reader (Amersham, Pharmacia Biotech, Piscataway, NJ, USA) and the Liscap Capture Application ver. 1.0. Electrophoretic gel analysis was performed using GelScan ver. 1.45 (Kucharczyk Electrophoretic Techniques, Warsaw, Poland). The plants identified as having a *Grande-1* retrotransposon fragment were used for genomic in situ hybridization (GISH).

## Mitotic chromosome preparations

Cytogenetic preparations were made according to the procedure described by Jenkins and Hasterok [42]. The root tips were excised from the 3-day-old seedlings, incubated in ice-cold water for 24 h and fixed in Carnoy's fixative (methanol and glacial acetic acid at a 3:1 ratio). The material was stored at -20 ˚C. To make cytogenetic preparations, the root meristems were washed in 0.01 M citrate buffer (pH 4.8) for 3 × 5 minutes, digested in an enzyme mixture containing 6% pectinase (v/v, Sigma), 1% cellulase (w/v, Sigma) and 1% cellulase 'Onozuka R-10'(w/v, Serva, Heidelberg, Germany) in citrate buffer at 37 ˚C for 2 h. The root tips were washed again in the citrate buffer. The meristems dissected from the root tips were placed in a drop of 45% acetic acid on a microscopic slide, covered with a coverslip, and gently squashed. Slides were incubated on dry ice to remove the coverslips and then stored at 4 ˚C until used.

## Probe labeling and genomic in situ hybridization

In order to discriminate maize chromosomes in oat genomic background, the maize *cv* 'Waza' total genomic DNA (gDNA) was used as a probe for genomic in situ hybridization (GISH). Maize gDNA was extracted from young leaves using standard C-TAB procedure and labeled with digoxygenin-11-dUTP (Roche, Basel, Switzerland) by nick-translation method according to manufacturer instructions.

GISH procedure followed the protocol described in Skrzypek et al. [22]. Briefly, the slides were pre-treated with RNase (100 μg/mL) in 2 × saline sodium citrate (SSC) buffer for 1 h, washed 3 × 5 min in 2 × SSC, post-fixed in 1% formaldehyde, washed again in 2 × SSC, and dehydrated in ethanol series (70%, 90%, 100%). The labeled maize gDNA was precipitated and dissolved in a hybridization mixture containing 50% deionized formamide, 10% dextran sulfate, 2 × SSC, 0.5% SDS, and water. The hybridization mixture was predenatured (75 ˚C, 10 min), applied to air-dried slides, and denatured again at 75 ˚C for 4.5 min. Hybridization was carried out in a humid chamber for about 40 hours at 37 ˚C. Post-hybridization washes were

performed in 10% formamide in 0.1 × SSC at 42 ˚C. The hybridization signals were detected by antidigoxigenin fluorescein-conjugated antibodies (Roche, Basel, Switzerland). The chromosomes were mounted and counterstained in VectaShield Antifade (Vector Laboratories, Burlingame, CA, USA) containing 2.5 mg/mL DAPI (Serva, Heidelberg, Germany). Hybridization signals were visualized and captured using fluorescence microscope Axio Imager Z2 (ZEISS, Germany) equipped with monochromatic camera AxioCam MRm (ZEISS, Oberkochen, Germany). The acquired images were digitally processed and superimposed using ZEN blue program (ZEISS, Oberkochen, Germany) and Photoshop CS3 (Adobe, San Jose, CA, USA).

## Maize chromosome identification

For maize chromosome identification, the total genomic DNA of OMA I and OMA II lines, oat *cv*. Bingo, and maize *cv*. Waza was extracted from ca. 0.8 g of fresh tissue frozen with liquid nitrogen using DNeasy Plant Mini Kit (Qiagen, Hilden, Germany). The added maize chromosomes in the tested OMA lines were identified by PCR with the simple sequence repeat (SSR) markers specific for particular maize chromosomes. Suitable SSR markers were chosen from the Maize Genome Database (https://www.maizegdb.org) and described before by Kynast et al. [9].

## Anatomical analysis of the leaves

To examine the anatomical arrangements, leaf material was harvested from the widest points of the fully expanded fourth leaves and prepared for light microscopy. Images of leaves' transverse cross-sections were carried out using the paraffin method as described by Ruzin [43]. Samples were fixed in 50% ethanol and next dehydrated in a graded series of ethanol. The tissues embedded in paraffin (Histowax, 56–58˚C) were cut using a sliding microtome (Leica SM 2000 R).The 5–7 μm thick sections were mounted on glass slides and double-stained with safranin (1%, w/v) and alcian blue (1%, w/v) then mounted in Canada balsam. The cross-sections were observed with the light microscope Leica DM2000 and photographed with the digital camera DFC320. The following morphometric analyses were made by using the IM 1000 software package: distance between two small vascular bundles (Fig 6A); total area of mesophyll (area of chlorenchyma within the leaf) between two small vascular bundles (Fig 6B); number of cells in inner bundle sheath (first ring of cells surrounding one small vascular bundle) (Fig 6C); number of cells in outer bundle sheath (second ring of cells surrounding one small vascular bundle) (Fig 6C); area of inner bundle sheath (total area of all cells); and area of outer bundle sheath (total area of all cells). The average size of a single cell in the inner bundle sheath was calculated based on the data of the area of the inner bundle sheath and the number of cells in the inner bundle sheath. The same procedure was done for the outer bundle sheath.

## Chlorophyll a fluorescence kinetics parameters

The measurements of (PCF) were conducted using a Handy PEA fluorometer (Hansatech, Kings Lynn, UK). The measurements were carried out on the flag leaves as described by Juzoń et al. [18]. The leaves' fragments were shaded with clips for about 20 minutes and then the measurement was taken. It was initiated by illuminating the leaf fragment with saturated light (intensity of 90 units = 5,400 μmol (quantum) $m^{-2}s^{-1}$, illumination time 0.9 s) to determine the maximum fluorescence ($F_m$) and calculate the maximum photochemical potential ($F_v/F_m$). Then, actinic light was switched on automatically (25 units = 1,500 μmol (quantum) $m^{-2}s^{-1}$) for a period of 270 s. When it was switched off, the leaf light was emitted by a diode emitting light in the far red range of about 15 W $m^{-2}$. On the basis of chlorophyll *a* fluorescence

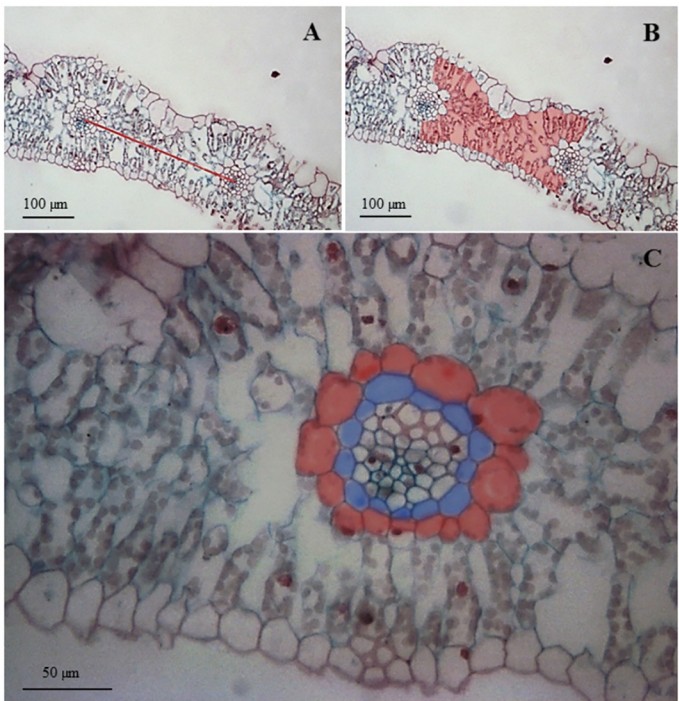

**Fig 6.** Morphometric analyses of oat (*Avena sativa* L.) leaf made by using the IM 1000 software package: (A) distance between two small vascular bundles; (B) total area of mesophyll (area of chlorenchyma within the leaf) between two small vascular bundles; (C) number of cells in inner bundle sheath (first ring of cells which surround one small vascular bundle–blue color) and number of cells in outer bundle sheath (second ring of cells which surround one small vascular bundle–red color).

measurements, the following parameters were calculated and analyzed: $F_v/F_m$–maximum photochemical efficiency of PSII; Area–area over the chlorophyll *a* fluorescence induction curve; PI–overall performance index of PSII photochemistry; ABS/CS–light energy absorption; $TR_0/CS$–excitation energy trapped in PSII reaction centers; $ET_0/CS$–energy used for electron transport; $DI_0/CS$–energy dissipated from PSII; and $RC/CS_0$ –number of active reaction centers in the excited leaf fragment.

## Yield component analyses

To estimate the impact of maize chromosomes added to oat genome, morphological observations (flag leaf length and width, height of main shoot, panicle length) were carried out and selected yield components (number of shoots/plant, number and weight of grains/plant, thousand grain weight (TGW)) were assessed.

## Statistical analysis

The results presented in the Figures constitute the mean values based on five plants as replicates. In each plant, the measurements were performed on five leaves. Means of data from leaf anatomical traits, parameters of chlorophyll *a* fluorescence kinetics, and yield components show S1 Table. One-way analysis of variance and Duncan's multiple range test at the 0.05 probability level was used to determine the significance of differences between lines/cultivar, which was marked with different letters. The linkage among observed yield components and chlorophyll *a* parameters were assessed utilizing Pearson's linear correlation coefficients. The

results were also analyzed by principal component analysis (PCA) to show them in multivariate approach. Data were analyzed using STATISTICA v. 13 (Stat-Soft, USA) software.

## Supporting information

**S1 Raw image. Original, uncropped electrophoresis gel picture and genomic in situ hybridization (GISH) underlying Fig 1 from the main text.** Identification of maize (*Zea maize* L.) chromatin added to the oat (*Avena sativa* L.) genome by PCR and genomic *in situ* hybridization (GISH). (A) The agarose gel with bands representing DNA fragments *Grande 1* (500 bp) specific for maize; path 1 –marker leader, path 2 –maize cv. Waza, path 3 –oat cv. Bingo, paths 4–13 DH lines of oat, paths 14–15 OMA lines, path 16 DH line of oat, X—lanes not included in the final figure. (B) Chromosomes of doubled haploid line (DH I), (C) Chromosomes of oat × maize addition line (OMA I). Blue fluorescence: DAPI, green fluorescence: maize genomic DNA (gDNA).
(PDF)

**S1 Table. Means of data from measurements of leaf anatomical traits, parameters of chlorophyll *a* fluorescence kinetics, and yield components.**
(XLS)

## Acknowledgments

Radenko Radošević for technical help to prepare transverse cross-sections of the leaves.

## Author Contributions

**Conceptualization:** Marzena Warchoł, Katarzyna Juzoń-Sikora, Dragana Rančić, Edyta Skrzypek.

**Formal analysis:** Marzena Warchoł, Katarzyna Juzoń-Sikora.

**Investigation:** Katarzyna Juzoń-Sikora, Dragana Rančić, Ilinka Pećinar, Tomasz Warzecha, Kamila Laskoś, Ilona Czyczyło-Mysza, Kinga Dziurka, Edyta Skrzypek.

**Methodology:** Marzena Warchoł, Dragana Rančić, Ilinka Pećinar, Tomasz Warzecha, Dominika Idziak-Helmcke, Edyta Skrzypek.

**Project administration:** Edyta Skrzypek.

**Resources:** Marzena Warchoł, Dragana Rančić.

**Supervision:** Marzena Warchoł.

**Visualization:** Dragana Rančić, Tomasz Warzecha, Dominika Idziak-Helmcke, Edyta Skrzypek.

**Writing – original draft:** Marzena Warchoł, Katarzyna Juzoń-Sikora, Edyta Skrzypek.

**Writing – review & editing:** Dragana Rančić, Ilinka Pećinar, Tomasz Warzecha, Dominika Idziak-Helmcke, Kamila Laskoś.

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
