## [Decision Letter · Decision Letter 0]

29 Aug 2023

PONE-D-23-25096 Comparative characteristics of oat doubled haploids and oat × maize addition lines: anatomical features of the leaves, chlorophyll a fluorescence and yield parametersPLOS ONE

Dear Dr. Warchol,

Thank you for submitting your manuscript to PLOS ONE. After careful consideration, we feel that it has merit but does not fully meet PLOS ONE’s publication criteria as it currently stands. Therefore, we invite you to submit a revised version of the manuscript that addresses the points raised during the review process.

We look forward to receiving your revised manuscript.

Kind regards,

Arun Kumar Shanker

Academic Editor

PLOS ONE

Journal Requirements:

https://journals.plos.org/plosone/s/file?id=ba62/PLOSOne_formatting_sample_title_authors_affiliations.pdf"

https://www.mdpi.com/2077-0472/13/2/243

https://www.ncbi.nlm.nih.gov/pmc/articles/PMC2948986/

In your revision ensure you cite all your sources (including your own works), and quote or rephrase any duplicated text outside the methods section. Further consideration is dependent on these concerns being addressed.

Additional Editor Comments:

The authors have to revise the MS according to reviewer1's suggestions so that the MS can be improved

Reviewers' comments:

Reviewer's Responses to Questions

**Comments to the Author**

1. Is the manuscript technically sound, and do the data support the conclusions?

Reviewer #1: Partly

Reviewer #2: Yes

2. Has the statistical analysis been performed appropriately and rigorously? 

Reviewer #1: No

Reviewer #2: Yes

3. Have the authors made all data underlying the findings in their manuscript fully available?

Reviewer #1: No

Reviewer #2: Yes

4. Is the manuscript presented in an intelligible fashion and written in standard English?

Reviewer #1: Yes

Reviewer #2: Yes

5. Review Comments to the Author

Reviewer #1: Manuscript "Comparative characteristics of oat doubled haploids and oat × maize addition lines: anatomical features of the leaves, chlorophyll a fluorescence and yield parameters" is very interesting.

General comments:

Authors evaluated how the presence of maize chromosomes changes the anatomical parameters of the leaves and functioning of photosynthetic apparatus of the disomic OMA lines compared to the DH lines and Bingo cultivar. Authors assessed also yield components for all lines.

Detailed comments:

The introduction is quite interesting. However, most of the references are very old.

Table 2 shows the results of the one-way ANOVA. I suggest supplementing is with mean squares for the difference factor and for the error. Then the table will be more readable.

Table 3: The title says that the table should include SE. Unfortunately, these values are missing from the table. It should be completed.

Table 4 shows the results of the one-way ANOVA. I suggest supplementing is with mean squares for the difference factor and for the error. Then the table will be more readable.

Table 5 shows the results of the one-way ANOVA. I suggest supplementing is with mean squares for the difference factor and for the error. Then the table will be more readable.

Figure 3: The caption says that the drawing should include SE. Unfortunately, these values are missing from the drawing. It should be completed.

Figure 4: The caption says that the drawing should include SE. Unfortunately, these values are missing from the drawing. It should be completed.

The description of statistical methods is cursory. Among other things, information about the empirical distribution and meeting the assumptions of analysis of variance is missing.

My suggestions:

The Authors evaluated yield components for all lines. Unfortunately, they were not tempted to analyze the effect of yield components on yield. This is a very important aspect in breeding. Such an analysis would have increased the value of the manuscript. I suggest conducting such an analysis.

In the Introduction, the authors wrote that, to the best of their knowledge, this is the first time that OMA and DH lines produced from the same parental crosses have been compared in terms of leaf anatomical characteristics. This would suggest a comparison between groups of OMA and DH lines. Unfortunately, such a comparison is missing from the manuscript. I suggest making such a comparison using, for example, contrast analysis.

The manuscript provides a description of many features. Each was discussed separately. In plant breeding, it is very important to determine the relationship between traits. This is very important in the selection process. I suggest conducting a correlation analysis.

Depending on the existence or absence of trait correlation, the selection process in breeding proceeds in different ways. The results of multivariate analysis are undoubtedly facilitating selection decisions. Supplementing the manuscript with the results of the analysis of canonical variables, together with the estimation of Mahalanobis distances between the studied genotypes, will be a valuable enrichment of the conducted research.

Paper needs major revision.

Reviewer #2: Dear Authors,

thank you for a lovely paper that neatly describes the study you have done. I enjoyed reading it. I have made some very minor language edits to the paper which I hope will contribute positively.

The file is attached for your use.

6. PLOS authors have the option to publish the peer review history of their article (what does this mean?). If published, this will include your full peer review and any attached files.

Reviewer #1: **Yes: **Jan Bocianowski

Reviewer #2: No

---

## [Author Response · Author response to Decision Letter 0]

24 Oct 2023

Dear Editor,

we would like to express our gratitude to the Editor and Reviewers for comments and critics which encourage us to improve our manuscript, especially statistical methods. Please find below our response to Reviewers’ comments since we would like to clarify some issues concerned by Reviewers. 

Journal Requirements:

Answer: We checked the requirements to conform the manuscript to proper form.

https://www.mdpi.com/2077-0472/13/2/243

https://www.ncbi.nlm.nih.gov/pmc/articles/PMC2948986/

In your revision ensure you cite all your sources (including your own works), and quote or rephrase any duplicated text outside the methods section. Further consideration is dependent on these concerns being addressed.

Answer: We rephrased duplicated text from Warzecha et al. 2023 (https://www.mdpi.com/2077-0472/13/2/243).

From Wesrhoff and Gowik 2010 (https://www.ncbi.nlm.nih.gov/pmc/articles/PMC2948986), we rephrased part about density of vein spacing in Kranz anatomy, and additionally we have corrected the error in the quoted literature. In our manuscript we cited Ueno and Sentoku (2006), whereas there should be Ueno et al. (2006).

Answer: We added the original gel image to Supporting Information (S1_Fig), one-line title and information in our manuscript. 

 S1 Fig. Original, uncropped electrophoresis gel picture underlying Fig 1 from the main text.

Answer: We checked the reference list and corrected citation in the text. 

We retracted one reference: Ueni O, Sentoku N. Comparison of leaf structure and photosynthetic characteristics of C3 and C4 Alloteropsis semialata subspecies. Plant, Cell Environ. 2006;29: 257–268. doi: 10.1111/j.1365-3040.2005.01418.x

We added 3 references:

Ueno O, Kawano Y, Wakayama M, Takeda T. Leaf vascular systems in C3 and C4 grasses: a two-dimensional analysis. Ann Bot. 2006; 97: 611–621. doi: 10.1093/aob/mcl010

Ishii T., Tanaka H., Eltayeb A.E., Tsujimoto H. (2013) Wide hybridization between oat and pearl millet belonging to different subfamilies of Poaceae. Plant Reprod., 26:25–32

Warzecha, T.; Bocianowski, J.; Warchoł, M.; Bathelt, R.; Sutkowska, A.; Skrzypek, E. Effect of Soil Drought Stress on Selected Biochemical Parameters and Yield of Oat × Maize Addition (OMA) Lines. Int. J. Mol. Sci. 2023, 24, 13905. https://doi.org/10.3390/ijms241813905

Reviewers' comments:

5. Review Comments to the Author

Reviewer #1: Manuscript "Comparative characteristics of oat doubled haploids and oat × maize addition lines: anatomical features of the leaves, chlorophyll a fluorescence and yield parameters" is very interesting.

General comments:

Authors evaluated how the presence of maize chromosomes changes the anatomical parameters of the leaves and functioning of photosynthetic apparatus of the disomic OMA lines compared to the DH lines and Bingo cultivar. Authors assessed also yield components for all lines.

Detailed comments:

R1: The introduction is quite interesting. However, most of the references are very old.

Answer: Best to our knowledge, we cited all previously published references on the characteristics of OMA lines in relation to anatomical studies, kinetics of chlorophyll a fluorescence and their yield.

However, we have added very recent information to the introduction regarding this topic published by: Warzecha, T.; Bocianowski, J.; Warchoł, M.; Bathelt, R.; Sutkowska, A.; Skrzypek, E. Effect of Soil Drought Stress on Selected Biochemical Parameters and Yield of Oat × Maize Addition (OMA) Lines. Int. J. Mol. Sci. 2023, 24, 13905. https://doi.org/10.3390/ijms241813905

R1: Table 2 shows the results of the one-way ANOVA. I suggest supplementing is with mean squares for the difference factor and for the error. Then the table will be more readable.

Answer: We corrected Table 2 as recommended by the reviewer. We added sum of squares, degrees of freedom, mean square and F-test values.

R1: Table 3: The title says that the table should include SE. Unfortunately, these values are missing from the table. It should be completed.

Answer: We completed Table 3 with SE values.

R1: Table 4 shows the results of the one-way ANOVA. I suggest supplementing is with mean squares for the difference factor and for the error. Then the table will be more readable.

Answer: We corrected Table 4 as recommended by the reviewer. We added sum of squares, degrees of freedom, mean square and F-test values.

R1: Table 5 shows the results of the one-way ANOVA. I suggest supplementing is with mean squares for the difference factor and for the error. Then the table will be more readable.

Answer: We corrected Table 5 as recommended by the reviewer. We added sum of squares, degrees of freedom, mean square and F-test values.

R1: Figure 3: The caption says that the drawing should include SE. Unfortunately, these values are missing from the drawing. It should be completed.

Answer: We completed Figure 3 with SE values.

R1: Figure 4: The caption says that the drawing should include SE. Unfortunately, these values are missing from the drawing. It should be completed.

Answer: We completed Figure 4 with SE values.

R1: The description of statistical methods is cursory. Among other things, information about the empirical distribution and meeting the assumptions of analysis of variance is missing.

Answer: We supplemented the paragraph with additional statistical methods: Pearson’s linear correlation coefficients and principal component analysis which were used while the manuscript was revised. We also change the structure of this paragraph.

R1: My suggestions:

The Authors evaluated yield components for all lines. Unfortunately, they were not tempted to analyze the effect of yield components on yield. This is a very important aspect in breeding. Such an analysis would have increased the value of the manuscript. I suggest conducting such an analysis.

Answer: We performed the Pearson’s linear correlation coefficients to find the connection among yield and certain yield elements and described it in the last part of Results. 

R1: In the Introduction, the authors wrote that, to the best of their knowledge, this is the first time that OMA and DH lines produced from the same parental crosses have been compared in terms of leaf anatomical characteristics. This would suggest a comparison between groups of OMA and DH lines. Unfortunately, such a comparison is missing from the manuscript. I suggest making such a comparison using, for example, contrast analysis.

Answer: The comparison of leaf anatomical traits of DH and OMA lines and cv. Bingo is presented in Table 3, were we applied the Duncan’s multiple test to compare DH and OMA lines. We also did analysis which revealed that there were no correlations of leaf anatomical traits between tested plant material (OMA and DH lines) with chlorophyll a fluorescence parameters and yields components. So, we do not present in the manuscript this analysis since it didn’t show any correlations.

R1: The manuscript provides a description of many features. Each was discussed separately. In plant breeding, it is very important to determine the relationship between traits. This is very important in the selection process. I suggest conducting a correlation analysis.

Answer: Thank you for the suggestion. We performed the Pearson’s linear correlation coefficients to determine the relationship between traits and we described that in Results sections.

R1: Depending on the existence or absence of trait correlation, the selection process in breeding proceeds in different ways. The results of multivariate analysis are undoubtedly facilitating selection decisions. Supplementing the manuscript with the results of the analysis of canonical variables, together with the estimation of Mahalanobis distances between the studied genotypes, will be a valuable enrichment of the conducted research.

Answer: We did not performed the analysis of canonical variables, together with the estimation of Mahalanobis distances between the studied genotypes because we do not have the access to such software. Instead of that we did Pearson’s linear correlation coefficients and biplot based on first two principal components axes (PC1 and PC2) for chlorophyll a fluorescence parameters of oat cv. Bingo, OMA and DH lines, distribution of oat genotypes based on the chlorophyll fluorescence parameters, for yield components of oat cv. Bingo, OMA and DH lines and distribution of oat genotypes based on the yield components. Instead of canonical variables, together with the estimation of Mahalanobis distances we performed the analysis of data with multivariate approach with the application of PCA analysis.

Reviewer #2: 

R2: Dear Authors,

thank you for a lovely paper that neatly describes the study you have done. I enjoyed reading it. I have made some very minor language edits to the paper which I hope will contribute positively.

The file is attached for your use.

Answer: Thank you for the language editing, we corrected the manuscript according to your suggestions. 

R2: This is ambiguous - the same number of chromosomes as in one or the same chromosome as in both have Chr9. the sentence could be clarified by saying "both OMA contained a single copy of maize chromosome 9"

Answer: We would like to explain the question connected with numbers of maize chromosomes: “In M&M section Line 347 we wrote that ...'and two OMA lines (Flamingstern × Bingo 347 and STH 9787(b) × Bingo) with the same number of maize chromosomes added to oat genome”. It means that in both OMA lines we detected 2 chromosomes. Chromosomes identification we showed in Table 1, where OMA I have 2 copy of maize chromosome 5 and OMA II have 1 copy of maize chromosome 3 and 1 copy of maize chromosome 8. We added that information to the M&M section.

---

## [Decision Letter · Decision Letter 1]

18 Jan 2024

Comparative characteristics of oat doubled haploids and oat × maize addition lines: anatomical features of the leaves, chlorophyll a fluorescence and yield parameters

PONE-D-23-25096R1

Dear Dr. Warchol,

We’re pleased to inform you that your manuscript has been judged scientifically suitable for publication and will be formally accepted for publication once it meets all outstanding technical requirements.

Kind regards,

Mayank Gururani

Academic Editor

PLOS ONE

Additional Editor Comments (optional):

Reviewers' comments:

Reviewer's Responses to Questions

**Comments to the Author**

1. If the authors have adequately addressed your comments raised in a previous round of review and you feel that this manuscript is now acceptable for publication, you may indicate that here to bypass the “Comments to the Author” section, enter your conflict of interest statement in the “Confidential to Editor” section, and submit your "Accept" recommendation.

Reviewer #1: All comments have been addressed

Reviewer #3: All comments have been addressed

2. Is the manuscript technically sound, and do the data support the conclusions?

Reviewer #1: Yes

Reviewer #3: Yes

3. Has the statistical analysis been performed appropriately and rigorously? 

Reviewer #1: Yes

Reviewer #3: Yes

4. Have the authors made all data underlying the findings in their manuscript fully available?

Reviewer #1: No

Reviewer #3: Yes

5. Is the manuscript presented in an intelligible fashion and written in standard English?

Reviewer #1: Yes

Reviewer #3: Yes

6. Review Comments to the Author

Reviewer #1: The Authors have taken all my comments into account when improving the manuscript. I recommend publishing the manuscript in its current form.

Reviewer #3: (No Response)

7. PLOS authors have the option to publish the peer review history of their article (what does this mean?). If published, this will include your full peer review and any attached files.

Reviewer #1: **Yes: **Jan Bocianowski

Reviewer #3: No

---

## [Editor Report · Acceptance letter]

29 Mar 2024

PONE-D-23-25096R1 

PLOS ONE

Dear Dr. Warchoł, 

I'm pleased to inform you that your manuscript has been deemed suitable for publication in PLOS ONE. Congratulations! Your manuscript is now being handed over to our production team.

Kind regards, 

on behalf of

Dr. Mayank Gururani 

Academic Editor

PLOS ONE